# AhmedML: High-Fidelity Computational Fluid Dynamics Dataset for Incompressible, Low-Speed Bluff Body Aerodynamics

## Abstract

The development of Machine Learning (ML) methods for Computational Fluid Dynamics (CFD) is currently limited by the lack of openly available training data. This paper presents a new open-source dataset comprising of high fidelity, scale-resolving CFD simulations of 500 geometric variations of the Ahmed Car Body - a simplified car-like shape that exhibits many of the flow topologies that are present on bluff bodies such as road vehicles. The dataset contains simulation results that exhibit a broad set of fundamental flow physics such as geometry and pressure-induced flow separation as well as 3D vortical structures. Each variation of the Ahmed car body were run using a high-fidelity, time-accurate, hybrid Reynolds-Averaged Navier-Stokes (RANS) - Large-Eddy Simulation (LES) turbulence modelling approach using the open-source CFD code OpenFOAM. The dataset contains boundary, volume, geometry, and time-averaged forces/moments in widely used open-source formats. In addition, the OpenFOAM case setup is provided so that others can reproduce or extend the dataset. This represents to the authors knowledge, the first open-source large-scale dataset using high-fidelity CFD methods for the widely used Ahmed car body that is available to freely download with a permissive license (CC-BY-SA).

## Introduction

The past decade has seen major efforts to improve the accuracy and computational efficiency of CFD approaches by utilizing the latest High-Performance Computing (HPC) hardware (e.g., GPUs, arm64-based processors (3; 24; 68; 20)) as well as designing new algorithms and methods to match this emerging new HPC hardware (51; 41; 26; 73; 70; 59). Alongside this code and solver development, there has also been considerable work to explore different turbulence modelling approaches that can achieve greater accuracy.g., wall-modeled Large-Eddy-Simulation (WMLES) (39; 15), Hybrid RANS-LES (10; 19; 66; 57; 56; 8; 54) using a range of underlying numerical approaches.

This combination of higher fidelity methods coupled with advances in HPC has enabled CFD methods to correlate better to experimental data for cases such as complete aircraft simulations (8; 39; 27) or realistic road cars (34; 5). However, whilst these advances have improved simulation accuracy, lowered run times and often reduced the cost of a CFD simulation, there still exists a fundamental link between these three factors. A coarser mesh or time-step may help to reduce the runtime or cost, but it would most likely worsen accuracy due to insufficient spatial or temporal resolution. Likewise to improve accuracy, the additional mesh or temporal resolution will increase the computational cost. This link is particularly important when it comes to developing tools that can provide real-time predictions, as reducing the prediction time to seconds with state of the art methods ends up worsening the accuracy or requiring very large number of HPC resources, which ultimately increases the cost.

Machine Learning (ML) methods, in particular, surrogate models, have emerged as a potential method (71; 16; 49; 72) that once trained can provide a rapid prediction tool at a time and cost much lower than traditional methods (2; 37; 11). However, ML methods available in the literature today, do not directly improve the accuracy compared to traditional CFD methods, so there still exists a need for traditional CFD approaches where high accuracy is needed, as well as the tool to provide the underlying training data.

The field of surrogate models via machine learning is broad but in general, two main approaches have emerged: physics-driven and data-driven (49). Physics-driven approaches (58; 32; 62; 47; 61) typically add the Partial Differential Equation (PDE) e.g Navier-Stokes, as a soft constraint or regularizer to the loss function in Physics-Informed Neural Networks (PINNs) (61) and Physics-Informed Neural Operators (PINOs) (47). Data-driven approaches on the other hand typically use machine learning techniques to learn the solutions or solution operator of PDEs (22), e.g., program the ML model in a supervised manner to minimize the difference between the 'true' solution and the predicted solution. The 'true' solution can become available by using traditional solvers or by experimental measurements. Examples of these data-driven approaches that are trained on supervised simulation data include message-passing graph neural networks (GNNs) (14; 25), e.g., MeshGraphNets (60; 23; 42) and pure data-driven Neural Operators (43; 44; 29; 45).

THE NEED FOR HIGH QUALITY TRAINING DATA AND RELATED WORK

The training dataset is a key ingredient to the development and validation of any ML method, whether they are data-driven or physics-driven. To date, there are orders of magnitude less open-source CFD data than in other communities such as natural language processing (NLP) and computer vision (18). Recently, several groups have generated large-scale training data to demonstrate the potential of their ML methods (e.g Jacob et al. (35), Li et al. (46), Baque et al. (12)) however none were made open-source and freely available, and therefore cannot be used by the community to develop their own methods. More recently, Bonnet et al. (13) released the AirfRANS dataset that includes more than 1000 2D airfoils simulated using a steady-state RANS setup using OpenFOAM, that is freely available to download under a permissive open-source license. This dataset is useful for the initial development of ML models, however there are numerous limitations including the 2D nature of the geometries, the steady-state lower fidelity method (RANS) and the large gap in complexity between this and a 3D plane or car. To address these limitations, Elrefaie et al. (21) developed the DrivAerNet dataset that includes 4000 geometric variants of the DrivAer geometry, also simulated using a steady-state RANS formulation, using OpenFOAM on meshes of between 8 and 16 million cells. This dataset has the benefits of the realism of a road-car but lacks the high-fidelity CFD simulation approach i.e steady-state RANS have been shown in numerous studies to not be accurate for large-scale bluff body flows (5; 6; 4) and the mesh sizes of 8-16M cells are an order of magnitude lower than typically used within industry (addressed by our upcoming DrivAerML dataset (9)). Finally the license that DrivAerNet dataset is provided under (CC-BY-NC) does not permit the use for commercial methods which limits it's use by the emerging eco-system of startups, software providers and tech companies.

FOCUS OF THIS WORK

In this work (and the associated WindsorML (7) and DrivAerML datasets (9), that were created in tandem with this AhmedML dataset) we focus on addressing the limitations of this previous work by firstly using higher-fidelity scale-resolving methods (WMLES or Hybrid RANS-LES), that are the state of the art for the CFD community and provide better correlation to ground-truth experimental data (5; 6; 4). And secondly, making the data freely available, in a consistent data format that is common across each dataset, and available under a permissive license (CC-BY-SA). Choosing meshes and CFD approaches that are closer to what industry would use for the chosen test-case is key, because a method that works for a 2D airfoil with 500k nodes (13) may not work for a 3D 20M cell case - due to both the underlying computational efficiency of the method and/or a limitation in the method itself e.g., MeshGraphNet (60) vs. multi-scale MeshGraphNet (23) that may not show differences at small mesh sizes but would for larger.

In addition, one of the main interests for ML-enhanced CFD is whether these models can provide a cheaper and faster prediction tool - thus testing it on realistically sized datasets is important to also accurately measure the time to train and do inference.

MAIN CONTRIBUTIONS

In this paper, we aim to address several of the aforementioned limitations of prior work to advance ML for CFD. This paper's novel contributions are summarized as follows:

- first openly available large-scale ML training dataset for the widely used Ahmed car body (1)
- the use of high-fidelity scale resolving CFD method which ensures the best possible correlation to experimental data and realism to real-life flow;
- comprehensive dataset including volume, boundary, geometry and forces & moments in open-source formats;
- inclusion of the OpenFOAM case setup to enable the community to reproduce and/or extend the dataset
- in combination with the WindsorML (7) and DrivAerML (9) provides three of the most widely used automotive bodies with consistent naming and access policies that together provide comprehensive datasets to test ML methods for CFD.
- open-source dataset (CC-BY-SA), that is free to download.

## TEST-CASE DESCRIPTION

### BACKGROUND

The Ahmed body (1) is a generic car geometry, shown in Figure 1; comprising a flat front with rounded corners and a sharp slanted rear upper surface. It has been subject to extensive wind-tunnel (48; 1; 53), as and CFD testing ((36; 50; 30; 52; 30; 31; 63; 17; 40; 33; 55; 63; 4)). This has been typically done for different slant back angles ($25^o$ and $35^o$) that represent both largely geometry and pressure-induced flow separation, together with 3D vortical structures. For that reason, this test case has proven popular amongst the CFD community (especially the turbulence modelling community), partly because of its car-like shape and because it maintains a challenging flow to predict even with higher-fidelity methods (4).

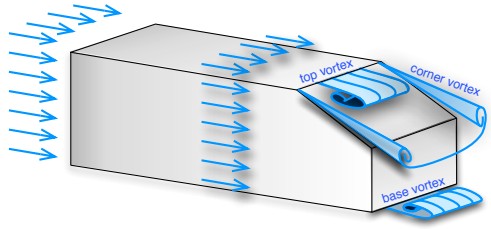

Figure 1: Schematic of the time-averaged flow structures over the baseline Ahmed Car body (4)

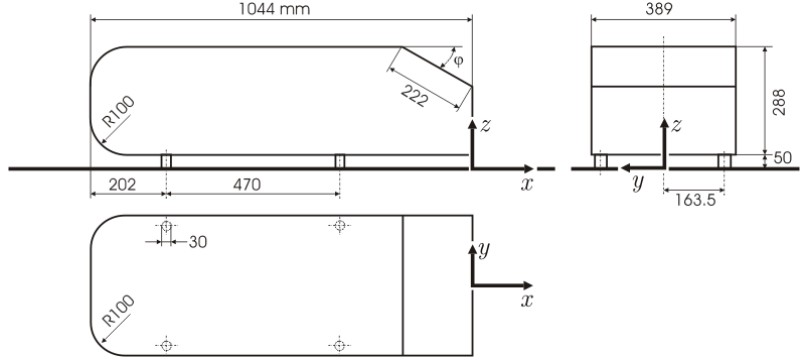

Figure 2: Dimensions of the baseline Ahmed car body

The Ahmed body provides many of the flow features found over road-cars, such as the large 3D separation region behind the car body and the roll up of vortices at the rear corners. The wake behind the car body is a mixture of both a highly turbulent separated flow as well as counter-rotating

Figure 3: Side view of the SnappyHexMesh generated mesh

vortices that interact with this wake. The angle of the rear slant is influential to the structure of the wake and the reattachment point. At $25^o$ the vortices have sufficient strength so that the flow is able to reattach half way down the slant due to the extra momentum from the vortices. At $35^o$ these counter-rotating vortices are weaker, which results in the flow being completely separated over the entire slant back of the vehicle (4). It is therefore a good baseline geometry to vary to create a large dataset of possible variations of the Ahmed car body that can exhibit many different flow features and which can therefore test the generalizability of an ML approach.

GEOMETRY

The geometry, as defined by Ahmed (1) is illustrated in Figure 2; the body has a length of $L = 1.044m$ with a height, $H = 0.288m$ and a reference area of $A_{ref} = 0.112$. In the experimental study, the body was mounted on four stilts, at a height of $0.05m$, to permit flow under the car which is important for 'ground effect'. To avoid additional mesh complexity, the stilts were here neglected, in line with previous numerical studies in the literature (4). Thus the body for the baseline geometry is fixed at the same height, $0.05m$, above the floor. We discuss the variations of this geometry in Section

BOUNDARY CONDITIONS

The flow is at a Reynolds number of $Re = 7.68 \times 10^5$ based on the body height $H$ (0.288m) (or $2.78 \times 10^6$ if based upon the body length (1.044m)), dynamic viscosity ($\nu = 3.75 \times 10^{-7}$) and the free-stream velocity $U_\infty = 1ms^{-1}$. At this Reynolds number the flow is largely turbulent and similar to those of a realistic road-car. An inlet condition is imposed $3m$ upstream of the body and an outlet condition is imposed $6m$ downstream. A no-slip wall condition is imposed on the ground floor and car body, with slip walls applied to the wind tunnel walls. The nutUSpaldingWallFunction (based upon the work of Spalding (67)) is used to model the near-wall given the high $y^+$ grid. The time step is set to $\Delta t U_\infty / L = 6 \times 10^{-4}$ (where $L$ is the length of the body), which ensures a maximum convective CFL number of less than one in areas of key flow separation. Each simulation was run for a total of 40 convective transit times ($=TU_\infty/L$); time-averaging began after the initial 10 transit times.

COMPUTATIONAL MESH

An unstructured mesh of approximately 15-20M prismatic and hex-dominant cells were used (depending on the input geometry) using the OpenFOAM internal meshing tool; snappyHexMesh (Figure 3). A high-$y^+$ approach was taken with 3 prism layers, resulting in a $y^+$ range of approximately 50 over the top surface of the body, which is exceeded at areas of significant flow acceleration and much lower where flow separation occurs. Uniform volumetric refinement was placed around the vehicle to reduce variation of the grid between geometry variations.

SOLVER SETUP

OpenFOAM v2212 was used for all the simulations, run in double-precision, compiled with OpenMPI 4.1. The pimpleFoam solver was used, which is a segregated, transient solver for incompressible, turbulent flow of Newtonian fluids. The Spalart-Allmaras based Delayed Detached-Eddy Simulation (SA-DDES) (65) model was used for all simulations in order to capture the large-scale geometry and pressure-induced flow separation. A hybrid 2nd-order upwind/central differencing spatial discretization scheme was used based upon the work of Travin et al. (69) for the momentum equations and a second-order upwind scheme for the turbulent variables. A second order upwind

scheme was used for temporal discretization. The Geometric AMG (GAMG) solver was used for the pressure variables and smoothSolver was used for the velocity and turbulent variables.

### HPC SETUP

All simulations were run on Amazon Web Services, using a dynamic HPC cluster provisioned by AWS ParallelCluster v3.9. Amazon EC2 hpc6a.48xlarge nodes were used for each case, which contain a dual-socket AMD Milan 96 core (in-total across both sockets) chip with 384Gb RAM. These are connected using a 100Gbit/s Elastic Fabric Adapter (EFA) interconnect (64). A 40TB Amazon Fsx for Lustre parallel file-system was used as the main data location during the runs, which were later transferred to object storage by Amazon S3. Each simulation was run on 288 cores using 4 hpc6a.48xlarge nodes that were selectively under-populated to 72 cores per node to obtain the optimum memory bandwidth. Each case took approximately 30mins to mesh and 48hrs to solve.

### VALIDATION

The primary aim of this work is to provide a dataset that can be used to develop and validate ML approaches, where the training data is based upon a CFD setup that can provide the best correlation to experimental data for the baseline geometry and have the best chance of also correlating to it's geometry variants. The CFD approach (i.e., meshing, turbulence model, numerical methods) used for this work represents the high fidelity approaches that would be used within academia and more importantly industry for bluff body shapes (5; 6; 4). As will be discussed in the next section, the Ahmed car body is a challenging test-case for CFD because the rear slant angle goes through an abrupt change in flow separation between $25^o$ and $35^o$ and thus only Direct Numerical Simulation (DNS) would have the potential to confidentially predict every geometric variant. Thus the validation within this paper is to show rigour and a sound methodology but ultimately it's not possible to achieve perfect correlation to the ultimate ground truth without extensive mesh sizes and computational expense that would provide diminishing returns.

### PRIOR WORK

The choice of a transient scale-resolving method i.e., Delayed Detached-Eddy simulation (DDES) (65) is based upon decades of work studying bluff body shapes, where RANS models has consistently failed to capture the correct flow physics (31; 38; 63; 28; 33; 55; 63). However in these works it was also shown that even though high-fidelity hybrid RANS-LES methods consistently provide better correlation to experimental data than lower-fidelity RANS methods, they still suffer from some inaccuracies. The source of these inaccuracies are mesh resolution, turbulence model formulation and numerical schemes used. Thus, the next section shows the validation work for our specific CFD approach for the baseline Ahmed car body geometry.

### RESULTS

We discuss here and compare against available experimental data the baseline Ahmed car body at $25^o$ rear-slant angle to justify the choice of CFD setup described in the prior section. As seen in Table 1, the correlation between the drag coefficient and lift coefficient is within 2% for the chosen resolution for the data (21.3M cells). It is clear that as the prior literature has shown, the correct prediction of this particular configuration of the Ahmed car body ($25^o$) is challenging and with successive grid refinement the solution has not reached complete convergence and indeed goes further away from exp. data. However, the fine mesh resolution was chosen as a compromise between computational cost/time and correlation to experimental data. Importantly the drag and lift are globally integrated values and thus it is more important to focus on the actual flow-field predictions.

It can be seen that for the fine mesh resolution, the overall correlation to the experimental data flow-field is good with the major flowfields captured (Figure 4 shows downstream visualizations for the streamwise velocity). The main discrepancy is in the initial shear-layer where the experimental data has a small separate bubble in the centerplane which reattaches half way down the rear slant of the body. This reduction of the centerline velocity is seen at all three planes in Figure 4 but this feature is missing in the CFD. However, the rest of the flow structures as well as the flow behind the

car body is well predicted and thus so is the overall lift and drag (Table 1). The validation of the approach against experimental data is discussed in greater detail in the Supplementary Information; SI.

Table 1: Force coefficients

| Source | Cd | Cl | Mesh Count (M) | Time-Step | CPU node-hours (hrs) |
|---|---|---|---|---|---|
| Exp. data (Meile et al. (53)) | 0.299 | 0.345 | | | |
| OpenFOAM (extra-fine) | 0.280 | 0.344 | 31.5 | $4{\times}10^{-4}$ | 498 |
| OpenFOAM (fine) (dataset) | 0.285 | 0.343 | 21.3 | $6{\times}10^{-4}$ | 184 |
| OpenFOAM (medium) | 0.293 | 0.328 | 10.1 | $8{\times}10^{-4}$ | 61 |
| OpenFOAM (coarse) | 0.303 | 0.347 | 3.5 | $1.2{\times}10^{-3}$ | 12 |

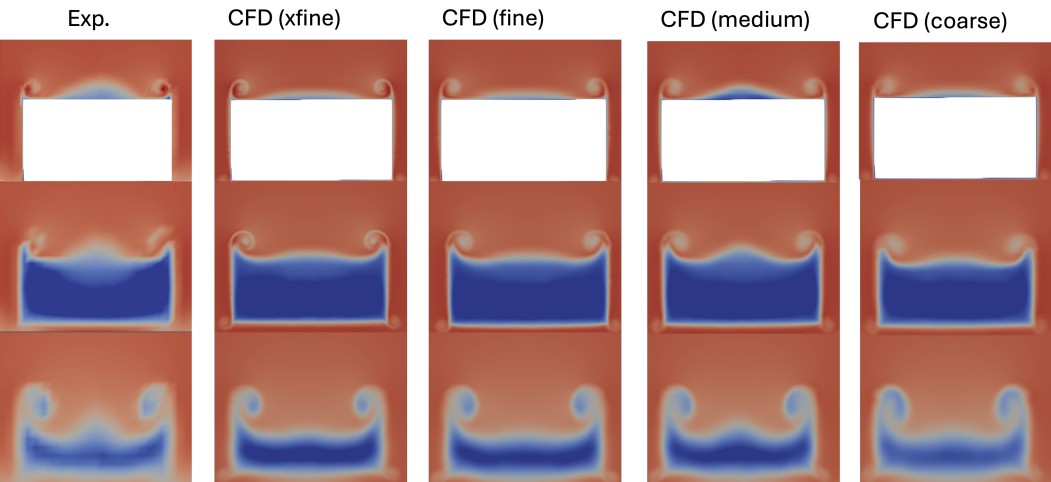

Figure 4: Streamwise (x) velocity comparison between experimental data (48) (left) and CFD (right) at planes (top to bottom) $x/H = 0, x/H = 0.27, x/H = 0.69$

## DATASET DESCRIPTION

### GEOMETRY VARIATIONS

The baseline geometry is adapted through 6 main parameters as shown in Table 7 that are sampled using Latin Hypercube to provide 500 geometries (see SI for further details). These were chosen to provide a suitable range of geometries that would exhibit different flow patterns e.g., pressure vs. geometry induced separation. For example, one of the parameters is the rear slant angle that as was previously discussed has been show in previous work to provide large changes in the overall flow separation as well as the corresponding lift/drag/moment coefficients that are of practical interest to the aerodynamics community. The min/max values of each value were based upon engineering judgement and to avoid completely unrealistic shapes. Figures 7d & 7c shows the wide range of lift and drag coefficients resulting from the geometry variants. Figures 7b & 7a illustrates the total pressure coefficient for a low and low drag result, which shows the large range of flow features that are produced in this dataset.

Table 2: Geometry variants of the Ahmed car body

| Part | | |
|---|---|---|
| Variable | Min | Max |
| angle of rear-window (degrees) | 10 | 60 |
| tilted surface length (mm) | 150 | 250 |
| length of the body (mm) | 800 | 1200 |
| height of the body (mm) | 250 | 315 |
| width of the body (mm) | 300 | 500 |
| radius of the front body (mm) | 80 | 120 |

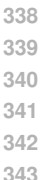
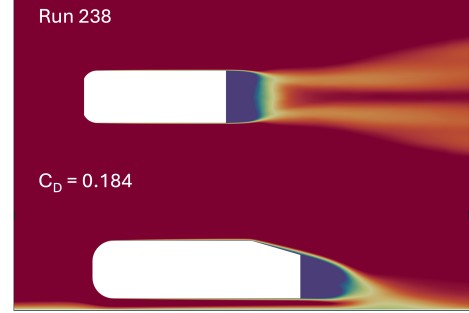

Run 205

$C_D = 0.508$

Run 238

$C_D = 0.184$

(a) Total Pressure Coefficient for high drag geometry variant example

(b) Total Pressure Coefficient for low drag geometry variant example

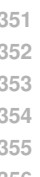
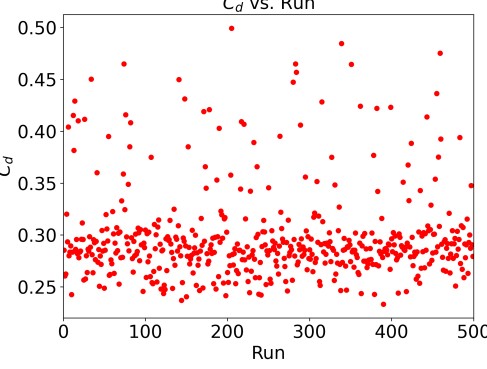
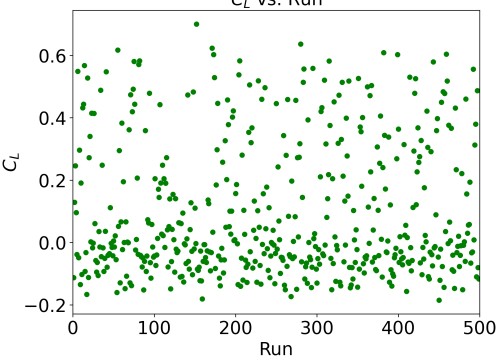

(c) Variation of drag coefficient against run number

(d) Variation of lift coefficient against run number

Figure 5: Variation of total pressure coefficient and force coefficients across a sample of the dataset

HOW TO ACCESS THE DATASET

The dataset is open-source under the CC-BY-SA license [1] and available to download via Amazon S3. In order to download, no AWS account is required and the full details in the dataset README [2] and SI.

DATASET CONTENTS

Table 5 summarizes the contents of the dataset (see SI for further details). All data within the current dataset is time-averaged (after an initial 10CTU's of flowtime to allow the flow to remove any initial transients). This was chosen for two reasons; firstly storing the instantaneous data (i.e each time-step) for every geometry variation would have increased the dataset size by $\approx$ x 80,000 (given the number

---

[1] https://creativecommons.org/licenses/by-sa/4.0/deed.en
[2] https://xxxxxxx.s3.us-east-1.amazonaws.com/ahmed/dataset/README.txt

Table 3: Summary of the dataset contents

| Output | Description |
|---|---|
| Per run (inside each run_i folder) | |
| ahmed_i.stl | surface mesh of the Ahmed car body geometry |
| boundary_i.vtp | time-averaged flow quantities (Pressure, Pressure Coefficient, Wall Shear Stress (vector), $y^+$) on the Ahmed car body surface |
| volume_i.vtu | time-averaged flow quantities within the domain volume |
| force_mom_i.csv | time-averaged drag & lift calculated using constant $A_{ref}$ |
| force_mom_varref_i.csv | time-averaged drag & lift calculated by case-dependant $A_{ref}$ |
| geo_parameters_i.csv | parameters that define the shape (in mm) |
| slices | folder containing .vtp 2D slices in X, Y,Z through the volume containing key flow-field variables |
| images | folder containing .png images of CpT and $U_x$ variables in X, Y, Z slices through the volume |
| Other | |
| force_mom_all.csv | time-averaged drag & lift for all runs using constant $A_{ref}$ |
| force_mom_varref_all.csv | time-averaged drag & lift for all runs using case-dependant $A_{ref}$ |
| geo_parameters_all.csv | parameters that define the shape for all runs |
| ahmedml.slvs | SolveSpace input file to create the parametric geometries |
| stl | folder containing stl files that were used as inputs to the OpenFOAM process |
| openfoam-casesetup.tgz | complete OpenFOAM setup that can be used to extend or reproduce the dataset |

of time-steps) and is not currently the norm for any engineering company for this reason. Secondly, for the practical use of CFD, time-averaged values are used because they represent what an object, such as a car, experiences during the course of it's operation and instantaneous values have much less practical interest.

## ML EVALUATION

We conducted an example ML evaluation using a Graph Neural Network (GNN) approach (more details in the SI) to demonstrate how this dataset could be used to train a ML model to predict unseen cases. We find that using a 60/20/20 split of train, validation and test, it is possible to obtain a MAE of less than 0.034 for the drag coefficient and MAE less than 0.085 for the lift coefficient via the integration of predicted wall-shear stress and pressure quantities on the 1.1M node surface mesh of the vehicles (shown in Figure 6).

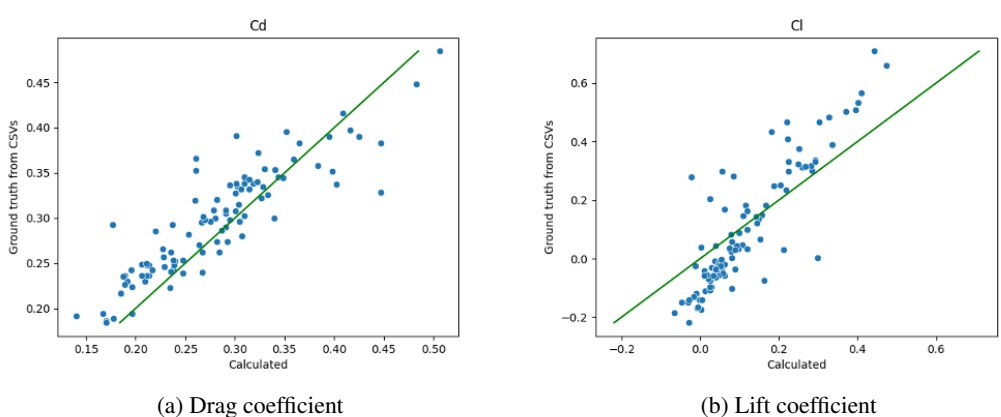

|  |  |
|:--:|:--:|
| (a) Drag coefficient | (b) Lift coefficient |

Figure 6: Actual vs predicted for the force coefficients obtained through integration of the wall-shear stress and pressure

LIMITATIONS OF THE DATASET

Whilst the dataset has numerous benefits over prior work e.g high-fidelity, large-scale, freely available it has several shortcomings:

- The dataset has purely geometrical differences with no variation in boundary conditions. Expanding the dataset to include boundary condition changes (i.e inflow velocity) would help ML developers to use this dataset for more than just geometry prediction.

- Whilst the Ahmed car body has become a standard test-case to assess CFD methods for bluff body separation (such as road car aerodynamics) it lacks the complexity of a real-life vehicle (addressed by the upcoming related DrivAerML dataset (9))

- The CFD modelling approach still does not exactly match experimental data and even higher-fidelity methods on finer meshes can yield even closer correlation to the experimental ground truth (e.g Direct Numerical Simulation (DNS)

- The dataset only includes time-averaged data rather than time-series. This is because industry typically focuses on time-averaged data can be be inspected for engineering usefulness. In addition the data requirements for capturing the full time-history (e.g 80,000 time-steps) which significantly increase the size of the dataset. However future work could be to include this for a limited number of runs to help develop models to capture the time-history.

CONCLUSIONS AND FUTURE WORK

This paper outlines the open-source AhmedML dataset that contains outputs from the Computational Fluid Dynamics simulations of 500 geometric variants of the Ahmed car body. These simulations were conducted using a high-fidelity, transient scale-resolving methodology in the open-source code; OpenFOAM. This comprehensive dataset includes volume and boundary data in commonly used open-source formats as well as time-averaged forces and the underlying geometry STLs. In addition 2D slices of the flowfield through $x, y$ & $z$ are provided as well as images of flow-field variables in those slice locations. This dataset alongside two further datasets released in tandem (WindsorML (7) & DrivAerML (9)) give developers and users of ML methods a rich source of data to further advance this important field, that holds the potential to offer significant runtime savings to both industry and academia.

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

CONTENTS

## A  NUMERICAL METHOD

The Navier-Stokes equations represent the fundamental motion of a fluid. They are derived from the conservation of mass and momentum, and can be expressed in Cartesian form for an incompressible Newtonian flow with constant density and variable viscosity (which are the assumptions used in the AhmedML dataset simulations) as follows:

$$\frac{\partial u_i}{\partial x_i} = 0 \tag{1}$$

$$\frac{\partial u_i}{\partial t} + u_j \frac{\partial u_i}{\partial x_j} = -\frac{1}{\rho}\frac{\partial p}{\partial x_i} + \frac{\partial \tau_{ij}}{\partial x_j} \tag{2}$$

where the viscous stress $\tau_{ij}$ is defined as:

$$\tau_{ij} = \mu \left( \frac{\partial u_i}{\partial x_j} + \frac{\partial u_j}{\partial x_i} \right) \tag{3}$$

While these equations fully represent any fluid flow, an analytical solution is only possible for basic flows that are subject to many assumptions. When an analytical solution is not possible, numerical methods may be used to achieve an approximate solution. In order to obtain a solution, these differential equations must be discretized to form integral equations which can be solved by a numerical procedure.

By solving these equations directly without further modelling or assumptions, the fluid velocity and pressure can be obtained, this method is called Direct Numerical Solution (DNS). Whilst it is the most complete numerical method as it involves no further assumptions or modelling, it is not practical for industrial applications using the current generation of computational resources. Three alternative approaches are discussed below; Reynolds Averaged Navier-Stokes (RANS), Large Eddy Simulation (LES) and hybrid RANS-LES - the final which is used to generate this dataset.

### A.1  REYNOLDS AVERAGED NAVIER-STOKES (RANS)

This method relies on providing a statistical representation of the turbulence via mathematical models, in which case none of the turbulence is resolved. All such methods are based on the Reynolds temporal decomposition; this involves applying an 'ensemble-averaging' process to the whole turbulent flow field, $\phi(x,t)$, which, for the case where the flow is statistically stationary, is given by:

$$\phi(x,t) = \Phi(x) + \phi^{'}(x,t) \tag{4}$$

where $\phi^{'}(x,t)$ is the fluctuating component and $\Phi(x)$ is the mean component, which can be defined as:

$$\Phi_i = \lim_{\tau \to \infty} \frac{1}{\tau} \int_{t_o}^{t_0+\tau} \phi_i \, dt \tag{5}$$

When this temporal decomposition is applied to the Navier-Stokes equations (Equations 1 and 2), the Reynolds-Averaged Navier-Stokes (RANS) equations are attained, which can be expressed as the following:

$$\frac{\partial U_i}{\partial x_i} = 0 \tag{6}$$

$$\frac{\partial U_i}{\partial t} + U_j \frac{\partial U_i}{\partial x_j} = -\frac{1}{\rho}\frac{\partial P}{\partial x_i} + \frac{1}{\rho}\frac{\tau_{ij}}{\partial x_j} - \frac{\partial \overline{u_i^{'} u_j^{'}}}{\partial x_j} \tag{7}$$

This averaging process produces an extra term that is not present in the Navier-Stokes equations. This term, $\overline{u_i^{'} u_j^{'}}$, is the Reynolds Stress tensor, which represents the effect of the turbulence fluctuations on

the mean flow. It leads to a further 6 variables and therefore modelling is required in order to obtain full closure of Equation 7. In the following, only incompressible flows are considered and as such $\rho$ is assumed to take a value of unity.

There are two main approaches to modelling the Reynolds Stresses. The first approach is to model each Reynolds Stress individually and solve a transport equation for each. This is the most complete method as individual components are accounted for independently and several terms of the transport equations are exact in their form. However as there are six Reynolds stresses, an additional six transport equations are required, as well as a transport equation for the length scale, such as $\varepsilon$ or $\omega$ which can increase the computational cost of a simulation.

An alternative approach to closing Equation 7 is to model these Reynolds Stresses using the Boussinesq approximation (4). This assumes that the effect of turbulence on the main flow can be treated phenomenologically as a viscosity, a 'turbulent viscosity'. This latter approach is now discussed in more detail, given this is the approach that serves as the baseline for the turbulence modelling method used in this work.

**Boussinesq approximation**    In the Navier-Stokes equations, the viscous stresses in the momentum equations are modelled by Newton's theory of viscosity. This states that the viscous stresses are linearly proportional to the mean rate of strain, $S_{ij}$. Boussinesq (4) proposed that the Reynolds stresses that result from the turbulent fluctuations in the velocity field can be modelled in a similar fashion, as they are shown to increase with the rate of deformation. It is logical to assume that as the molecular viscosity links the viscous stresses and strains, then the stresses due to turbulence (the Reynolds stresses) can be linked by a turbulent viscosity, $\nu_t$ to the mean strain rate. While physically, the molecular viscosity is a thermodynamic property of a fluid, the turbulent viscosity is dependent on the local flow itself, and this is only an engineering approximation rather than an actual physical property;

$$-\overline{u_i' u_j'} \;=\; 2\nu_t S_{ij} - \frac{2}{3}k\delta_{ij},\tag{8}$$

where

$$\delta_{ij} \;=\; \begin{cases}1, & \text{if } i = j \\ 0, & \text{if } i \neq j\end{cases}\tag{9}$$

$$k \;=\; \frac{1}{2}\overline{u_i' u_i'}\tag{10}$$

$$S_{ij} \;=\; \frac{1}{2}\left(\frac{\partial U_i}{\partial x_j} + \frac{\partial U_j}{\partial x_i}\right)\tag{11}$$

In order to solve the equations, the turbulent viscosity must be determined. There are numerous approaches (e.g Algebraic (16), 1-equation (17) (11), 2-equation (13) based upon variants of velocity and length scales) but one of the most widely is the Spalart-Allmaras model (20) which is based on a different modelling strategy, and is derived using dimensional analysis and a term-by-term modelling approach.

**Spalart-Allmaras RANS model**    The Spalart-Allmaras solves a transport equation for a term $\tilde{\nu}$, that is equal to the turbulent viscosity far from any walls:

$$\frac{D\tilde{\nu}}{Dt} = \overbrace{c_{b1}(1 - f_{t2})\tilde{S}\tilde{\nu}}^{\text{Source}} - \overbrace{\left[c_{w1}f_w - \frac{c_{b1}}{\kappa^2}f_{t2}\right]\left(\frac{\tilde{\nu}}{d}\right)^2}^{\text{Sink}} +$$

$$\underbrace{\frac{1}{\sigma}\left[\frac{\partial}{\partial x_j}\left((\nu + \tilde{\nu})\frac{\partial\tilde{\nu}}{\partial x_j}\right) + c_{b2}\frac{\partial\tilde{\nu}}{\partial x_i}\frac{\partial\tilde{\nu}}{\partial x_i}\right]}_{\text{Diffusion}}\tag{12}$$

Which is solved to calculate the turbulent viscosity:

$$\nu_t = f_{v1}\tilde{\nu} \tag{13}$$

Where,

$$f_{v1} = \frac{\chi^3}{\chi^3 + c_{v1}^3} \tag{14}$$

$$\chi = \frac{\tilde{\nu}}{\nu} \tag{15}$$

Where $\nu$ is the molecular kinematic viscosity, and additional terms are defined as follows:

$$\tilde{S} = \Omega + \frac{\tilde{\nu}}{\kappa^2 d^2} f_{v2} \tag{16}$$

In which $\Omega = \sqrt{2\Omega_{ij}\Omega_{ij}}$ is the vorticity magnitude, $d$ is the distance to the nearest wall and $\kappa = 0.41$ is the Von Karman constant. There are several damping functions introduced which are calibrated to match the experimentally observed variation of $\nu_t$ near the wall:

$$f_{v2} = 1 - \frac{\chi}{1 + \chi f_{v1}} \tag{17}$$

The damping function $f_w$ damps the sink term of Equation 12 near the wall and also provides a sensitivity to the pressure gradient.

$$f_w = g\left[\frac{1 + c_{w3}^6}{g^6 + c_{w3}^6}\right]^{1/6} \tag{18}$$

$$g = r + c_{w2}(r^6 - r) \tag{19}$$

$$r = \frac{\tilde{\nu}}{\tilde{S}\kappa^2 d^2} \tag{20}$$

The additional term $f_{t2}$ is designed to account for transition and enables to model 'trip' itself from laminar to turbulent flow:

$$f_{t2} = c_{t3}exp(-c_{t4}\chi^2) \tag{21}$$

The model constants are as follows:

| $c_{b1}$ | $c_{b2}$ | $\sigma$ | $c_{w1}$ 0.9 | $c_{w2}$ | $c_{w3}$ | $c_{v1}$ | $c_{t3}$ | $c_{t4}$ 0.9 |
|---|---|---|---|---|---|---|---|---|
| 0.1355 | 0.622 | 2/3 | $c_{b1}/\kappa^2 + (1 + c_{b2})/\sigma$ | 0.3 | 2 | 7.1 | 1.2 | 0.5 |

Table 4: Model coefficients for the SA model

The reader is advised to visit the NASA Spalart Allmaras website that has extensive details of all the published SA model versions (http://turbmodels.larc.nasa.gov/spalart.html).

A.2 Large Eddy Simulation (LES)

An alternative approach to solving the instantaneous Navier-Stokes equations is Large Eddy Simulation (LES), in which a filtering operation is conducted which separates the smallest turbulent scales from the largest. This is based upon the assumption that the large scales in a turbulent flow are those containing the most energy, are anisotropic and are dependant on the flow geometry and thus should be resolved. The small scales can be thought as being more isotropic, dissipative and not as dependant on the flow geometry, and as such can be modelled. This means the mesh can be coarser and thus allows a significant saving on mesh resolution compared to DNS.

The process of scale separation is achieved by applying a filter to the velocity field, where the instantaneous velocity field is split into a resolved and residual part:

$$u(x,t) = \widehat{U}(x,t) + u^{'}(x,t) \tag{22}$$

This filtering procedure can be achieved with different types of filters but in practice the maximum grid size is commonly used as the filter which is why the modelled component is typically called the 'sub-grid' component.

Applying this filter to the momentum equations leads to:

$$\frac{\partial \widehat{U}_i}{\partial t} + \frac{\partial \widehat{U_i U_j}}{\partial x_i} = -\frac{1}{\rho}\frac{\partial \widehat{p}}{\partial x_i} + \frac{\partial \widehat{\tau}_{ij}}{\partial x_i} \tag{23}$$

It is important to note that because the product of the filtered velocities is not the same as the filter of the product of the velocities:

$$\widehat{U_i U_j} \neq \widehat{U}_i\,\widehat{U}_j, \tag{24}$$

then a modelling approximation must be introduced:

$$\widehat{U_i U_j} = \widehat{U}_i\,\widehat{U}_j + \widehat{u^{'}_i u^{'}_j} \tag{25}$$

Applying this to Equation 23 gives:

$$\frac{\partial \widehat{U}_i}{\partial t} + \frac{\partial \widehat{U}_i\,\widehat{U}_j}{\partial x_i} = -\frac{1}{\rho}\frac{\partial \widehat{p}}{\partial x_i} + \frac{\partial \widehat{\tau}_{ij}}{\partial x_i} - \widehat{u^{'}_i u^{'}_j} \tag{26}$$

To close the equations, a model is required for the sub-grid scale stress, $\tau^s_{ij}$ using the filtered(resolved) velocities. A simple model based on a similar concept to the Bousinessq approximation was introduced by Smagorinsky (19):

$$-\widehat{u^{'}_i u^{'}_j} = 2\nu_{SGS}\widehat{S}_{ij} - \frac{2}{3}k_{SGS}\delta_{ij}, \tag{27}$$

The sub-grid scale viscosity, $\nu_{SGS}$ is modelled in a similar way to the mixing length model, by assuming it is proportional to the length scale and filtered strain. The length scale is calculated using the filter width $\Delta$ and $k_{SGS}$ is the sub-grid scale energy.

$$\nu_{SGS} = (C_s \Delta)^2\,\widehat{S^*} \tag{28}$$

Where $C_s$ is the Smagorinksy constant, $\widehat{S^*} = \sqrt{2\widehat{S}_{ij}\widehat{S}_{ij}}$, and the filter width is $\Delta = 2\left(\Delta_x \Delta_y \Delta_z\right)^{1/3}$. Lilly (12) assumed an inertial range specturm and deduced a value of $C_s = 0.17$, however Deadroff (5) found that in the presence of a wall the value should be reduced and typically values of $C_s = 0.065$ are used.

## A.3 HYBRID RANS-LES

The choice between RANS and LES methods is dependant on many factors, such as the accuracy required or the physical quantities that are of interest. For flows that are largely steady and for which only mean quantities are of interest (such as an airfoil at a low angle of attack), RANS modelling is often a suitable and cheap choice. However for complex flows that contain regions of large separation or strong anisotropy (such as the Ahmed body), then traditional RANS techniques often fail. For many cases their inability to capture the large scale unsteadiness and a tendency to under-predict the shear-stress in the separated layer (which leads to a longer separation length) mean they are unsuitable for these types of flows (9). Even more advanced Reynolds Stress models that are able to account for the anisotropy in the flow are often unable to provide an good enough description of the unsteady flow field for highly separated flows (8).

Whilst LES models can in general provide a much better alternative to RANS modelling for unsteady flows, they do so at a much higher cost, so much higher that for high-Reynolds numbers flow these costs are too great for general purpose calculations. As the Reynolds number increases, the size of the smallest eddies decrease and therefore so must the grid and time step to account for this. For many engineering problems in the aerospace and automobile industry the computational resources are simply too great.

An alternative approach that balances computational cost with accuracy are hybrid RANS-LES methods. These methods solve a RANS model in regions where the flow is attached, more simple to model and thus requires a coarser grid and then switch to a LES simulation in the regions where the RANS model would perform poorly such as the separation region behind the body. Arguably one of the first methods to pioneer this hybrid approach was Detached-Eddy Simulation (DES) (22). This approach is based on splitting the domain into RANS and LES regions based on the grid resolution. DES can be seen as a three-dimensional, unsteady model based on an underlying 'off-the-shelf' RANS model. It seamlessly acts as a sub-grid scale model in regions where the grid is fine enough to support a LES, and as a RANS model in areas where it is not. The main advantage of this approach is that it is conceptually easy to understand and mathematically simple because it only requires the modification of a single term. In DES, the RANS turbulent length scale is adapted to Equation 29 which effectively changes the turbulent length scale to one based on the grid resolution. For the original SA-DES model this results in every instance of the turbulent length scale being substituted for the new definition, as shown in Equation. 31.

$$L_{DES} = \min\left(L_{RANS}, L_{LES}\right) \tag{29}$$
$$L_{LES} = C_{DES}\Delta \tag{30}$$

Where $L_{RANS}$, $L_{LES}$ and $L_{DES}$ represent the RANS, LES and combined DES turbulent length scales respectively. $\Delta$ is the LES filter width which is typically taken as the cell volume for unstructured codes, although other expressions such as the maximum cell size are sometimes used (6). $C_{DES}$ is an empirical parameter that needs to be tuned to match the correct level of dissipation, typically $C_{DES} = 0.60$ but this constant is highly dependant on the numerics of the code and the underlying RANS model, and thus must be tuned for any new formulations.

$$\frac{D\tilde{\nu}}{Dt} = c_{b1}\tilde{S}\tilde{\nu} - \underbrace{c_{w1}f_w\left(\frac{\tilde{\nu}}{L_{DES}}\right)^2}_{\text{Mod}} + \frac{1}{\sigma}\left[\frac{\partial}{\partial x_j}\left((\nu + \tilde{\nu})\frac{\partial\tilde{\nu}}{\partial x_j}\right) + c_{b2}\frac{\partial\tilde{\nu}}{\partial x_i}\frac{\partial\tilde{\nu}}{\partial x_i}\right] \tag{31}$$

The original intention of DES (22) is to enable the boundary layers and attached steady flow to be modelled in RANS mode, i.e $L_{RANS} < L_{LES}$, and then for the separated unsteady flow away from the boundary layer to be resolved in LES mode, i.e $L_{LES} < L_{RANS}$. As the LES length scale is based on the grid resolution, this requires the mesh to be suitably designed to have a finer cells in the proposed LES region and coarser cells in the RANS regions.

**Delayed Detached-Eddy simulation (DDES)** Since the inception of DES, several groups found that a potential error was possible when the near-wall grid refinement was too great. This meant

that the model turned to its LES mode within the attached boundary layer which lowered the turbulent viscosity and for extreme cases triggered too early separation. This was termed grid-induced separation (GIS) (14) and was caused by the modelled turbulence level dropping but no resolved content was available to replace it (also often referred to as Modelled-Stress Depletion(MSD)).

In order to stop this from occurring, regardless of the mesh resolution in the boundary layer, a boundary layer 'shield' was integrated into the DES length scale equation to enforce the RANS mode in the attached boundary layers. This was originally achieved for the SST-DES model using the SST $F_1$ and $F_2$ limiters (15) but was later generalised for any RANS model, which became known as Delayed Detached-Eddy Simulation (DDES) (21) as shown below:

$$f_d = 1 - \tanh\left(\left[8r_d\right]^3\right), \tag{32}$$

where

$$r_d = \frac{\nu_t + \nu}{\sqrt{U_{i,j}U_{i,j}}\kappa^2 y^2}, \tag{33}$$

and, $\kappa$ is the Karman constant, and $y$ is the distance to the wall. $r_d$ equals 1 in the boundary layer and gradually reduces to 0 towards the edge of the boundary layer. The function $f_d$ is designed to be 1 in the LES region where $r_d \ll 1$, and 0 elsewhere.

This modification means that $L_{DDES}$ is now redefined to:

$$L_{DDES} = L_{RANS} - f_d \max\left(0, L_{RANS} - C_{DDES}\Delta\right) \tag{34}$$

This new DDES formulation was evaluated in the DESider (9) and ATAAC (18) projects and has now made the original DES version obsolete.

For this reason the turbulence model used within this work is the SA-DDES version (21) that has been widely used in industry and shown to give good correlation for bluff bodies (1).

## B  DATASET

### B.1  LICENSING TERMS

The dataset is provided with the Creative Commons CC-BY-SA v4.0 license[3]. A full description of the license terms is provided under the following URL:

```
https://xxxxx.s3.us-east-1.amazonaws.com/ahmed/dataset/LICENSE.txt
```

### B.2  ACCESS TO DATASET

The dataset is hosted on Amazon Web Services (AWS) via an Amazon S3 bucket

```
s3://xxxxx/ahmed/dataset
```

The dataset README.txt will be kept up to date for any changes to the dataset and can be found at the following URL:

```
https://xxxxx.s3.us-east-1.amazonaws.com/ahmed/dataset/README.txt
```

Please ensure you have enough local disk space before downloading (complete dataset is ~ 8 TB) and consider the examples below that provide ways to download just the files you need:

The first Step is to install the AWS Command Line Interface (CLI): `https://docs.aws.amazon.com/cli/latest/userguide/getting-started-install.html`.

The second Step is to use the AWS CLI to download the dataset. Follow the following examples for how to download all or part of the dataset.

Note 1 : If you don't have an AWS account you will need to add –no-sign-request within your AWS command i.e aws s3 cp –no-sign-request –recursive etc...
Note 2 : If you have an AWS account, please note the bucket is in us-east-1, so you will have the fastest download if you have your AWS service or EC2 instance running in us-east-1.

**Example 1: Download all files (~8 TB)**

```
aws s3 cp --recursive s3://xxxxx/ahmed/dataset .
```

**Example 2: Only download select files (STL, surface mesh & force):**

Create the following bash script that could be adapted to loop through only select runs or to change to download different files e.g boundary/volume:

```bash
#!/bin/bash

# Set the S3 bucket and prefix
S3_BUCKET="xxxxxx"
S3_PREFIX="ahmed/dataset"

# Set the local directory to download the files
LOCAL_DIR="./ahmed_data"

# Create the local directory if it doesn't exist
mkdir -p "$LOCAL_DIR"

# Loop through the run folders from 1 to 500
for i in $(seq 1 500); do
    RUN_DIR="run_$i"
    RUN_LOCAL_DIR="$LOCAL_DIR/$RUN_DIR"

    # Create the run directory if it doesn't exist
```

---

[3]`https://creativecommons.org/licenses/by-sa/4.0/deed.en`

```
1188        mkdir -p "$RUN_LOCAL_DIR"
1189
1190        # Download the ahmed_i.stl file
1191        aws s3 cp "s3://$S3_BUCKET/$S3_PREFIX/$RUN_DIR/ahmed_$i.stl" \
1192        "$RUN_LOCAL_DIR/" --only-show-errors
1193
1194        # Download the force_mom_i.csv file
1195        aws s3 cp "s3://$S3_BUCKET/$S3_PREFIX/$RUN_DIR/force_mom_$i.csv" \
1196        "$RUN_LOCAL_DIR/" --only-show-errors
1197        # Download surface meshes
1198        aws s3 cp  "s3://$S3_BUCKET/$S3_PREFIX/$RUN_DIR/boundary_$i.vtp" \
1199        "$RUN_LOCAL_DIR/" --only-show-errors
1200    done
```

### B.3 LONG-TERM HOSTING/MAINTENANCE PLAN

The data is hosted on Amazon S3 as it provides high durability, fast connectivity and is accessible without first requesting an account or credentials (only AWS CLI tools, described above, which are free to download and use). A dedicated website is currently being created for this dataset and the two associated datasets; WindsorML (2) and DrivAerML (3) datasets to help further clarify where the data is hosted and to communicate any additional mirroring sites.

### B.4 INTENDED USE & POTENTIAL IMPACT

The dataset was created with the following intended uses:

- Development and testing of data-driven or physics-driven ML surrogate models for the prediction of surface, volume and/or force coefficients

- Academic testing test-case given the ability to define the geometry, run the OpenFOAM case and train a model in a reproducable manner.

- As a potential benchmark test-case to assess the performance of different ML approaches for an open-source bluff-body dataset.

- Large-scale dataset to study bluff-body flow physics i.e formation of geometry vs pressure-induced separation and 3D vortical structures.

### B.5 DOI

At present there is no specific DOI for the dataset (only for this associated paper) - however the authors are investigating ways of assigning DOI to this Amazon S3 hosted dataset.

### B.6 DATASET CONTENTS

For each geometry there is a separate folder (e.g `run_1`, `run_2`, ..., `run_i`, etc.) , where "i" is the run number that ranges from 1 to 500. All run folders contain the same time-averaged data which is summarized in Table 5.

#### B.6.1 TIME-AVERAGING

The dataset only includes time-averaged data rather than time-series. This is because industry typically focuses on time-averaged data can be be inspected for engineering usefulness. In addition the data requirements for capturing the full time-history (e.g 80,000 time-steps) which significantly increase the size of the dataset. However there is on-going work to include a limited number of runs that contain the full time-history to help develop models to capture the time-history.

Table 5: Summary of the dataset contents

| Output | Description |
|---|---|
| Per run (inside each run_i folder) | |
| ahmed_i.stl | surface mesh of the Ahmed car body geometry (~ 150k cells) |
| boundary_i.vtp | time-averaged flow quantities (Pressure, Pressure Coefficient, Wall Shear Stress (vector), $y^+$) on the Ahmed car body surface (~ 1.1M cells) |
| volume_i.vtu | time-averaged flow quantities within the domain volume (~ 20M cells) |
| force_mom_i.csv | time-averaged drag & lift calculated using constant $A_{ref}$ |
| force_mom_varref_i.csv | time-averaged drag & lift calculated by case-dependant $A_{ref}$ |
| geo_parameters_i.csv | parameters that define the shape (in mm) |
| slices | folder containing .vtp 2D slices in $X, Y, Z$ through the volume containing key flow-field variables |
| images | folder containing .png images of CpT and $U_x$ variables in $X, Y, Z$ slices through the volume |
| Other | |
| force_mom_all.csv | time-averaged drag & lift for all runs using constant $A_{ref}$ |
| force_mom_varref_all.csv | time-averaged drag & lift for all runs using case-dependant $A_{ref}$ |
| geo_parameters_all.csv | parameters that define the shape for all runs |
| ahmedml.slvs | SolveSpace input file to create the parametric geometries |
| stl | folder containing stl files that were used as inputs to the OpenFOAM process |
| openfoam-casesetup.tgz | complete OpenFOAM setup that can be used to extend or reproduce the dataset |

### B.6.2 FORCE COEFFICIENTS

The drag and lift coefficients are defined as follows:

$$C_\text{D} = \frac{F_x}{0.5\,\rho_\infty\,|U_\infty|^2\,A}\,, \quad C_\text{L} = \frac{F_z}{0.5\,\rho_\infty\,|U_\infty|^2\,A}\,, \tag{35}$$

Where $F$ is the integrated force, $A$ is the frontal area of the geometry. The reference density $\rho_\infty$ is set to 1.

Two outputs are provided for the force coefficients; one in which the reference area is kept constant ($0.112\ \text{m}^2$) across all details (force_mom_i.csv), and secondly one where it based upon the frontal-area of each different geometry variant (force_mom_varref_i.csv).

### B.7 BOUNDARY DATA

Like the integrated forces the pressure field (`pMean`) is also given as dimensionless pressure coefficient $C_p$ (`static(p)_coeffMean`) defined as follows:

$$C_p = \frac{p - p_\text{ref}}{0.5\rho_\infty|U_\infty|^2}\,, \tag{36}$$

The reference density is set to 1, the reference pressure to 0 and the velocity is set to $1\text{m/s}$, to match the reference inflow velocity.

The wall friction coefficient $C_f$ can be computed from the wall shear stress vector (either per component or the magnitude), however in this dataset the wall-shear stress vector is given directly (`wallShearStressMean`), and users can compute the skin-friction themselves via the following formula:

$$C_f = \frac{\tau_w}{0.5\rho_\infty|U_\infty|^2}\,. \tag{37}$$

These two quantities (pressure and wall-shear stress) can then also be used to compute directly the force coefficients such as the lift and drag.

Finally, the $y^+$ value (non-dimensional wall distance) is given as:

$$y^+ = \frac{u_* y}{\nu} \tag{38}$$

where $u_*$ is the friction velocity at the wall, $y$ is the wall-distance and $\nu$ is the kinematic viscosity.

### B.8 VOLUME DATA

In addition to the static pressure coefficient previously defined, the total pressure coefficient $C_{pt}$ can be defined as follows:

$$C_{pt} = \frac{p_t - p_\text{ref}}{0.5\rho_\infty|U_\infty|^2}\,, \quad p_t = p + 0.5\rho_\infty|U_i|^2\,, \tag{39}$$

where the given definition for the total pressure $p_t$ is valid for an incompressible fluid.

### B.9 2D SLICES OF VOLUME SOLUTION

38 two-dimensional slices through the volume solution are provided. The $x$-normal slices range from $x = -1.3\,\text{m}$ to $x = 0.8\,\text{m}$ (with $0.1\,\text{m}$ intervals), the $y$-normal slices from $y = -0.4\,\text{m}$ to $y = 0.4\,\text{m}$ (with $0.1\,\text{m}$ intervals) and $z$-normal slices from $z = 0\,\text{m}$ to $z = 0.6\,\text{m}$, (with $0.1\,\text{m}$ intervals).

### B.10 IMAGES

In addition to the 2D slices, .png images of the Total Pressure Coefficient and Streamwise velocity are provided at the same points as the slice locations in $X$, $Y$, & $Z$ to either train ML models to directly predict these images or to quickly inspect the flow fields for each geometry. The legends have been removed to aid computer vision ML techniques but the legend values are from 0 to 1 for streamwise velocity and -1 to 1 for the total pressure coefficient.

### B.11 FILE FORMATS

All provided data is either written in ASCII or in the open source format VTK (i.e. *.vtp and *.vtu) to ensure the broadest compatibility.

Table 6: List of output quantities in the provided dataset files, all quantities are time-averaged.

| | | **volume_i.vtu** | |
|---|---|---|---|
| Symbol | Units | Field name | Description |
| $\overline{p^*}$ | $[\mathrm{m}^2/\mathrm{s}^2]$ | pMean | relative kinematic pressure |
| $\overline{U_i}$ | $[\mathrm{m/s}]$ | UMean | velocity vector |
| $\overline{u_i' u_j'}$ | $[\mathrm{m}^2/\mathrm{s}^2]$ | UPrime2Mean | resolved Reynolds stress tensor |
| $\overline{\nu_t}$ | $[\mathrm{m}^2/\mathrm{s}]$ | nutMean | turbulent eddy viscosity |
| $\overline{C_p}$ | $[-]$ | static(p)_coeffMean | static pressure coefficient |
| $\overline{C_{pt}}$ | $[-]$ | total(p)_coeffMean | total pressure coefficient |
| $\overline{\varpi}$ | $[1/\mathrm{s}^2]$ | vorticityMean | vorticity vector |
| | | **slices/<slice_<position>.vtp** | |
| Symbol | Units | Field name | Description |
| $\overline{p^*}$ | $[\mathrm{m}^2/\mathrm{s}^2]$ | pMean | relative kinematic pressure |
| $\overline{U_i}$ | $[\mathrm{m/s}]$ | UMean | velocity vector |
| $\overline{u_i' u_j'}$ | $[\mathrm{m}^2/\mathrm{s}^2]$ | UPrime2Mean | resolved Reynolds stress tensor |
| $\overline{\nu_t}$ | $[\mathrm{m}^2/\mathrm{s}]$ | nutMean | turbulent eddy viscosity |
| $\overline{C_p}$ | $[-]$ | static(p)_coeffMean | static pressure coefficient |
| $\overline{C_{pt}}$ | $[-]$ | total(p)_coeffMean | total pressure coefficient |
| $\overline{\varpi}$ | $[1/\mathrm{s}^2]$ | vorticityMean | vorticity vector |
| | | **boundary_i.vtp** | |
| Symbol | Unit | Field name | Description |
| $\overline{p^*}$ | $[\mathrm{m}^2/\mathrm{s}^2]$ | pMean | relative kinematic pressure |
| $\overline{y^+}$ | $[-]$ | yPlusMean | $y^+$ |
| $\overline{\tau_i}$ | $[\mathrm{m}^2/\mathrm{s}^2]$ | wallShearStressMean | wall shear stress vector |
| $\overline{C_p}$ | $[-]$ | static(p)_coeffMean | static pressure coefficient |

### B.12 OPENFOAM CASE SETUP

In addition to the outputs from the simulations, the original OpenFOAM case setup is provided. Please note that the outputs from the simulations have been renamed to meet the naming convention of the dataset, however the case is setup to produces all the required files e.g .vtu and .vtp files for the volume & boundary etc. The only step required for the user of the code is to install OpenFOAM v2212 and place the required .stl into the constant/triSurface/ folder.

## B.13 GEOMETRY VARIANTS

500 geometric variations of the Ahmed car body (as per Table 7) were created to produce a broad range of shapes that would produce differing flow physics e.g geometry-induced separation, pressure-induced separation and 3D vortical structures. The challenge then being for ML methods to be able to train and ultimately predict unseen examples across these broad dataset. Figures 8 & 9 show the Total Pressure Coefficient for the first 50 runs of the dataset at $y = 0$ and $z = 0$. They illustrate the broad range of flow separation patterns i.e complete flow separation from the rear surface, to a fully attached flow.

In order to efficiently create the range of geometries, the open-source parametric 3D CAD tool SolveSpace [4] was used. First, a base parametric model was created by hand in SolveSpace GUI according to Ahmed specifications and saved to file in the SolveSpace text format (.slvs). A custom python script randomised each parameter for each variant using Latin hypercube sampling before writing each variant as a separate .slvs text file. Next, using python script, for each variant the SolveSpace command-line executable was invoked to load the variant .slvs and its geometries exported as a triangulated mesh in STL format with 20,000 vertices and chord tolerance of 0.005mm.

To illustrate the large differences in flow physics, Figures 7a & 7b shows the total pressure coefficient for a low and low drag result, which shows the large range of flow features that are produced in this dataset i.e run205 has fully separated flow over the rear-window, whereas run238 has it fully attached. There is even more of a dramatic effect when looked in the $z$ slice, where the vortical structures create a much wider wake, which in part contributes to the much higher drag.

Table 7: Geometry variants of the Ahmed car body

| Part | | |
|---|---|---|
| Variable | Min | Max |
| angle of rear-window (degrees) | 10 | 60 |
| tilted surface length (mm) | 150 | 250 |
| length of the body (mm) | 800 | 1200 |
| height of the body (mm) | 250 | 315 |
| width of the body (mm) | 300 | 500 |
| radius of the front body (mm) | 80 | 120 |

---

[4]https://github.com/solvespace/solvespace

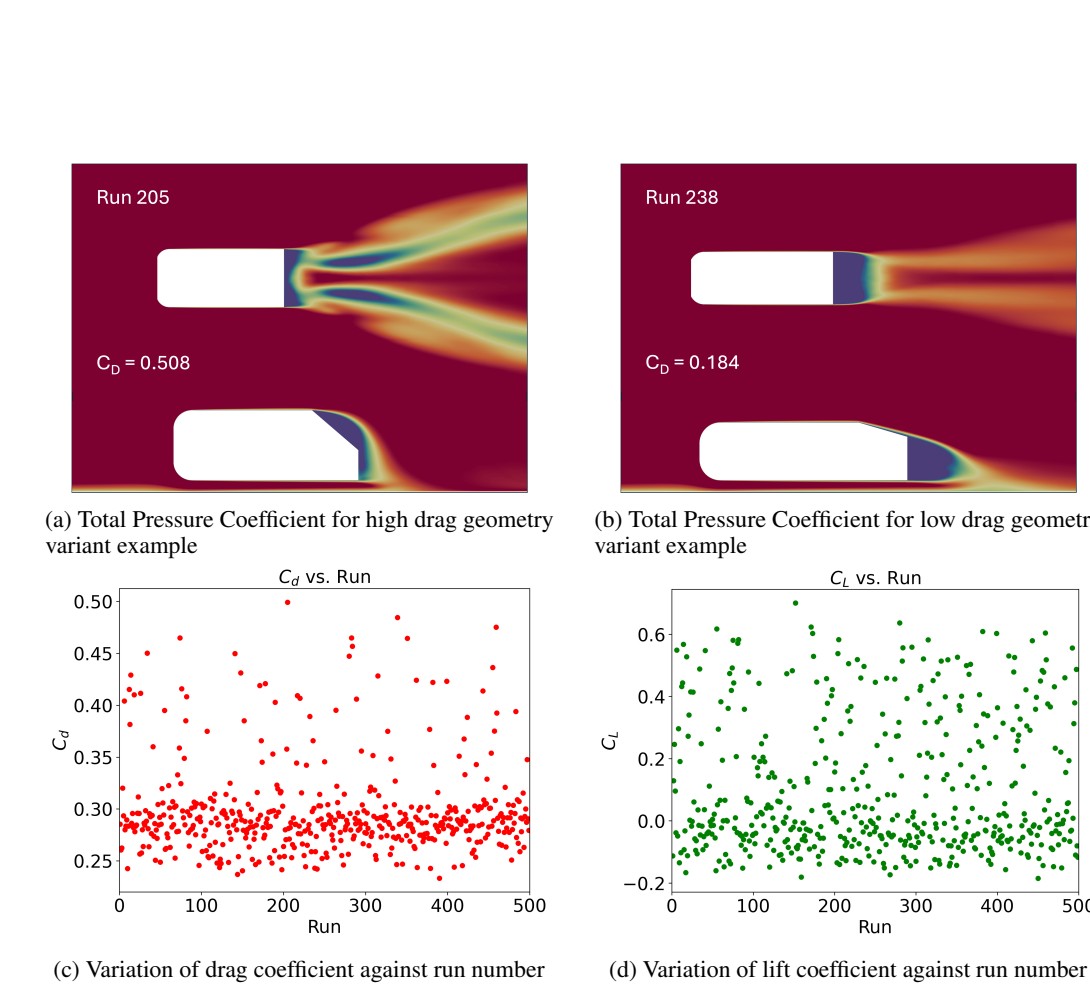

(a) Total Pressure Coefficient for high drag geometry variant example

(b) Total Pressure Coefficient for low drag geometry variant example

(c) Variation of drag coefficient against run number

(d) Variation of lift coefficient against run number

Figure 7: Variation of total pressure coefficient and force coefficients across a sample of the dataset

Figure 8: Total Pressure Coefficient at $y = 0$ for runs 1 to 50

Figure 9: Total Pressure Coefficient at $z = 0$ for runs 1 to 50

# C  ML EVALUATION

## C.1  SUMMARY

We have conducted preliminary analysis on our dataset using a modified version of one of the state-of-the-art scientific machine learning (SciML) methods, MeshGraphNet (7) on various tasks to illustrate the practicality of the dataset for ML evaluation. We utilize the encoder-processor-decoder architecture in MeshGraphNets and modify the method to enable it to make time-averaged predictions (2).

The entire Ahmed dataset is split into training (60%), validation (20%), and test (20%) sets. For each use case, the model is trained on the training set, and the checkpoint that had the best validation error was used to obtain the inference results on the test set.

For the Ahmed dataset, using the predicted surface pressure and wall-shear stress on the 1.1M node vtp surface mesh, we obtain predictions (shown in Figure 10) for the drag coefficient with a mean absolute percentage error (MAPE) of 0.101 and a mean absolute error (MAE): 0.034. For the lift coefficient the mean absolute error (MAE): 0.085. The surface contours of the actual, predicted and error for the mean pressure and wall-shear stress are shown in Figure 11. Training time is approximately 36 hours on x8 NVidia L40s GPUs and the inference time is less than a minute on the same hardware. Please note that these runs are preliminary and further work to optimize the methodology and hyperparameters is on-going which will published in future papers.

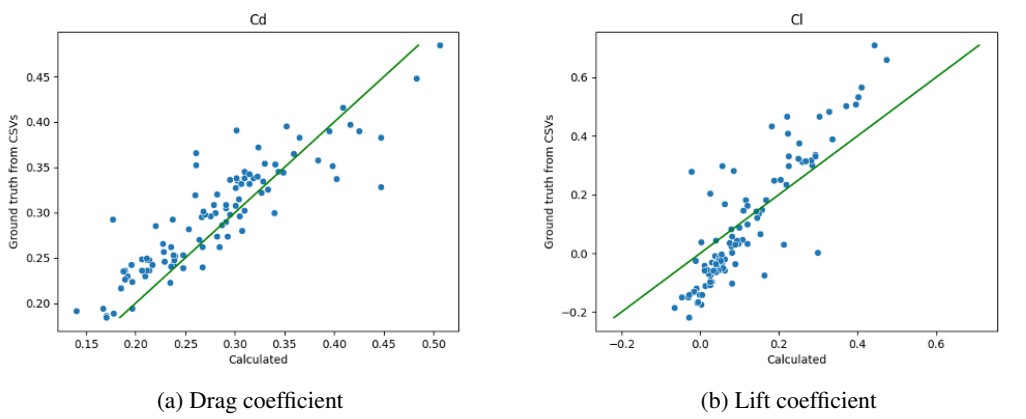

(a) Drag coefficient                                (b) Lift coefficient

Figure 10: Actual vs predicted for the force coefficients obtained through integration of the wall-shear stress and pressure

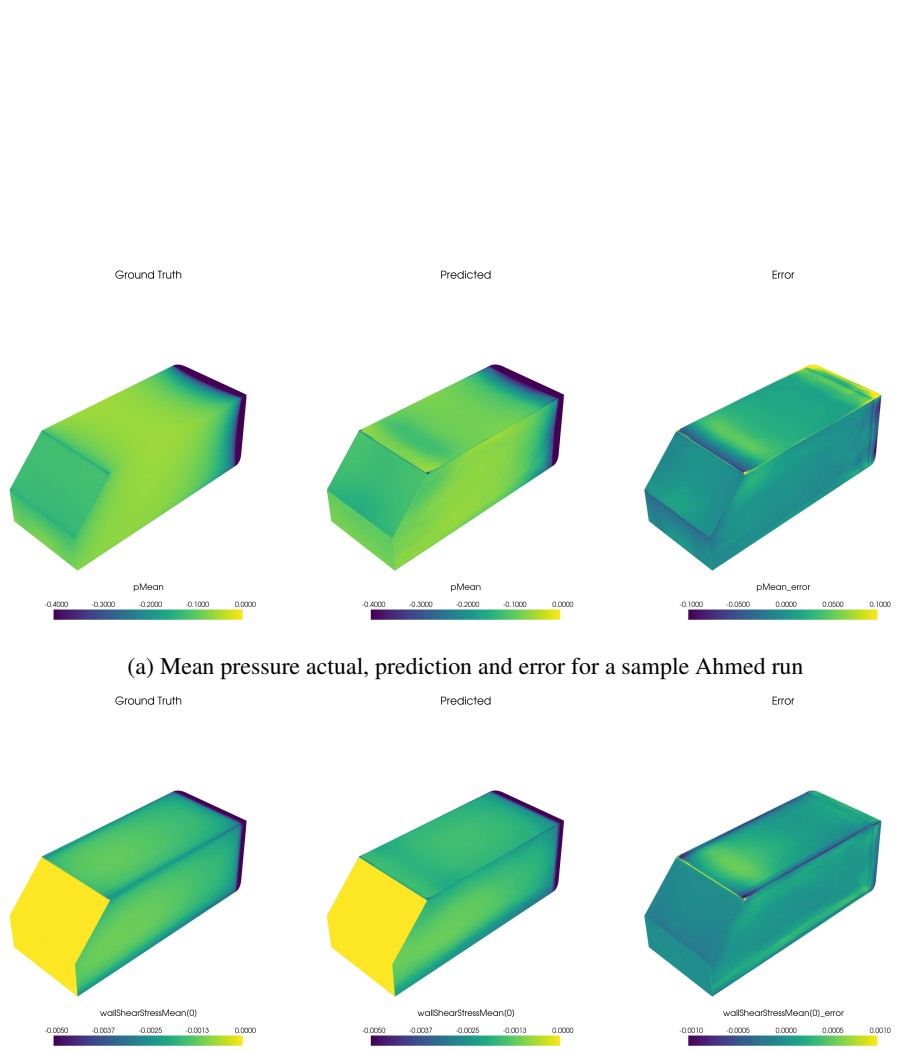

(a) Mean pressure actual, prediction and error for a sample Ahmed run

(b) Mean wall-shear stress actual, prediction and error for a sample Ahmed run

Figure 11: Actual, Prediction and error for the mean pressure and wall-shear stress for a sample Ahmed unseen geometry

# D ADDITIONAL VALIDATION DETAILS

This section provides additional information beyond what was discussed in the main body of the paper in Section 3.

## D.1 MESHES

Figure 13 shows three views of the unstructured mesh generated by SnappyHexMesh, the OpenFOAM mesh generation utility. The mesh design was choosen to cover all possible geometry variants and thus therefore potentially overrefined for some cases but reduces the risk of flow features from a geometry not being covered by enough mesh resolution. The choice of a high $y^+$ mesh was based upon prior work that shown little sensitivity between wall-resolved and high $y^+$ approaches for bluff body separation (10).

## D.2 ADDITIONAL POST-PROCESSING

As can be seen in Figure 14 for the baseline Ahmed car body, the flow is highly unsteady with both low and high frequency features. To ensure the time-averaged flow was sufficiently time-averaged, the flow was run for 80 convective time units (CTU) for all cases.

In addition the experimental data comparison shown in the main paper, Figures 12a that show that with the chose CFD method there is a slight inaccuracy in the initial separation at the beginning of the rear-window for the centerline position ($y = 0$) however in agreement with the planes behind the ahmed body, the velocity profiles show much closer agreement outside of this small area, which explains why the overall drag and lift coefficient is well predicted.

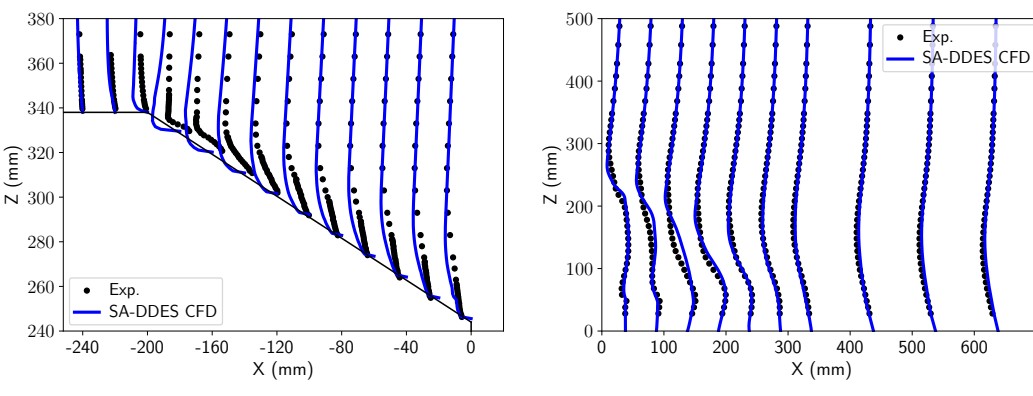

(a) Streamwise (x) velocity at $y = 0$ plane          (b) Streamwise (x) velocity behind the Ahmed car body

Figure 12: Mesh resolution

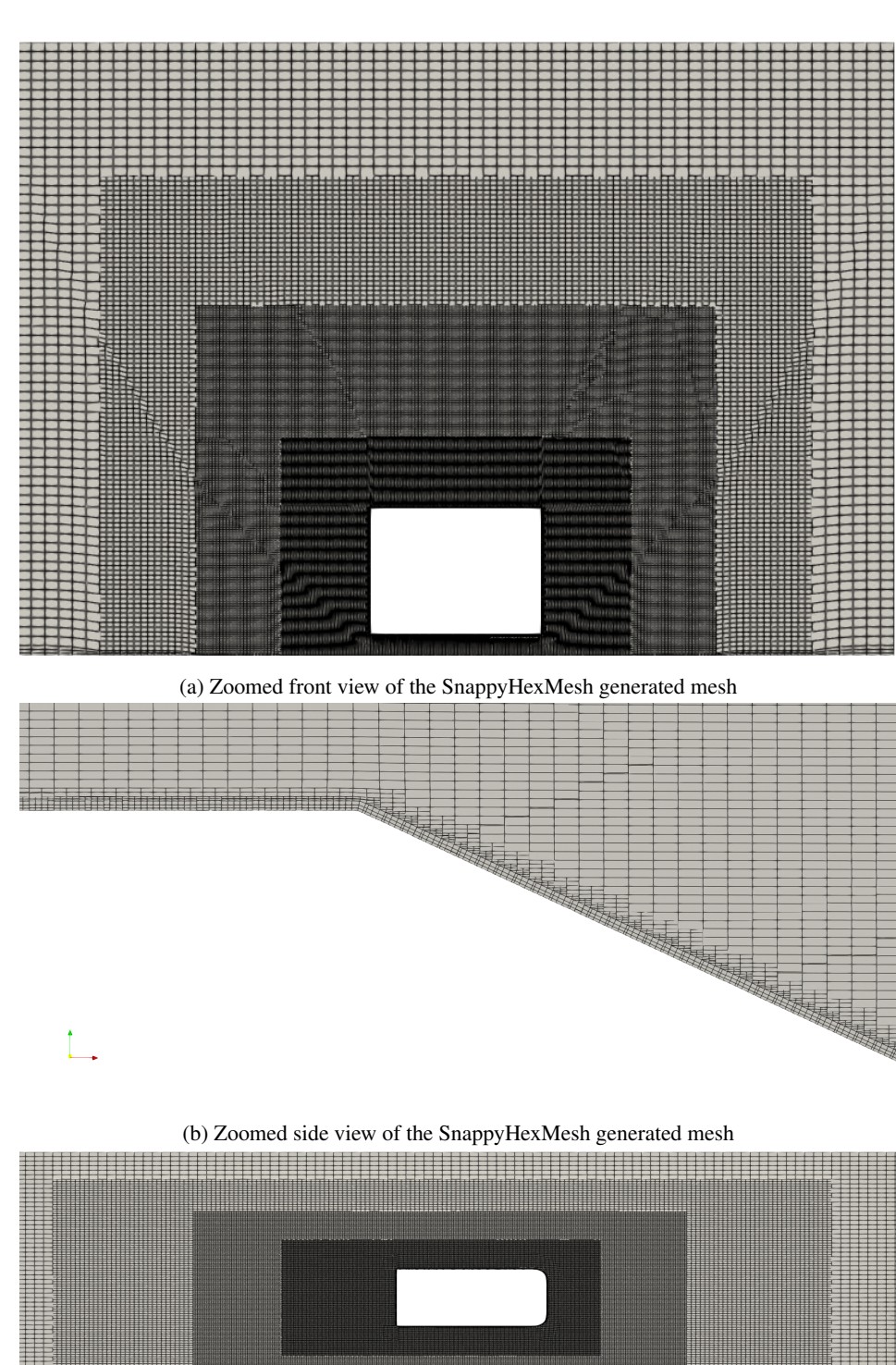

(a) Zoomed front view of the SnappyHexMesh generated mesh

(b) Zoomed side view of the SnappyHexMesh generated mesh

(c) Zoomed top view of the SnappyHexMesh generated mesh

Figure 13: Images of the unstructured mesh for the baseline Ahmed car body

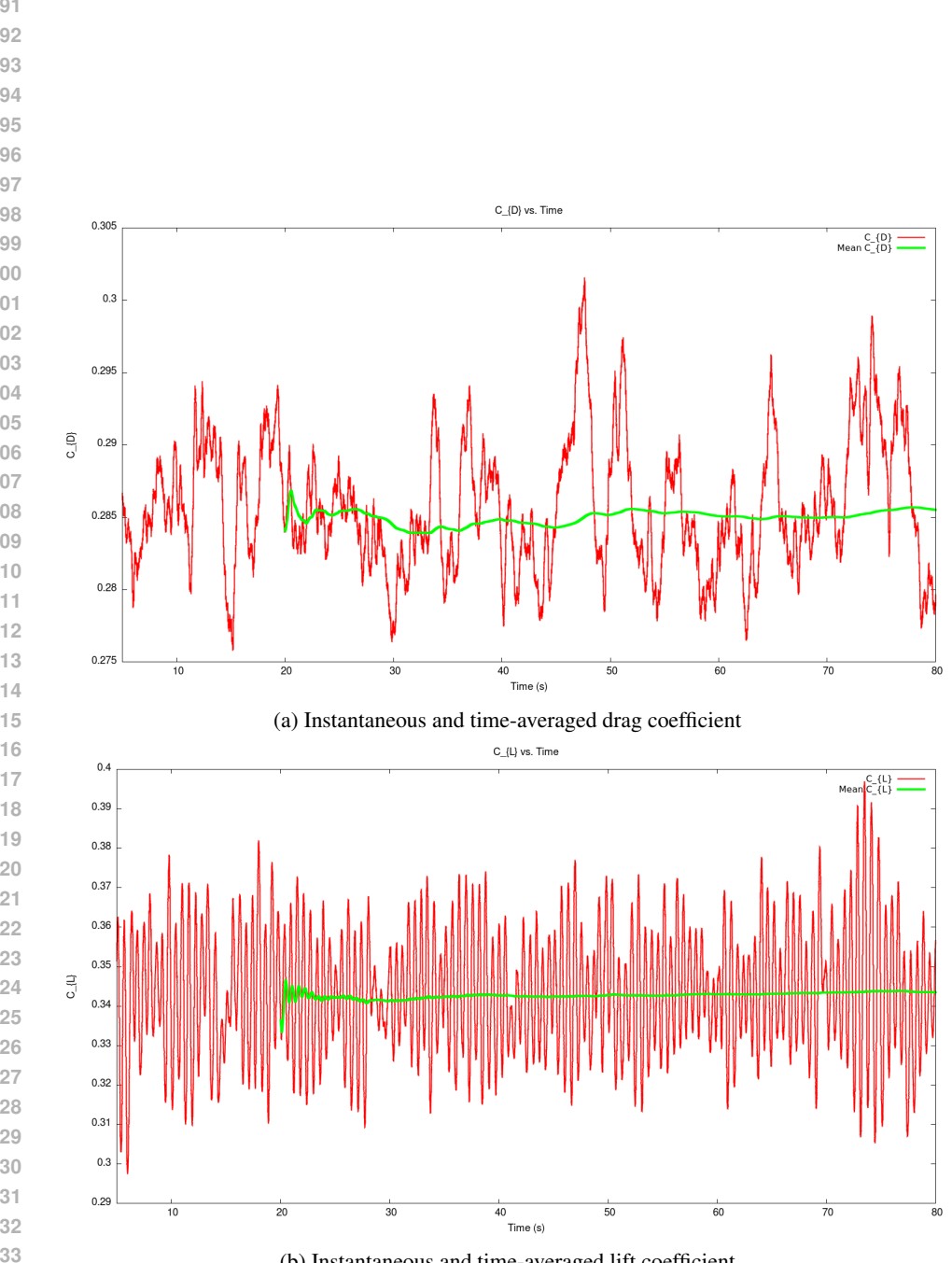

(a) Instantaneous and time-averaged drag coefficient

(b) Instantaneous and time-averaged lift coefficient

Figure 14: Instantaneous and time-averaged force coefficients to illustrate the need for 80 CTU's of time-averaging

# E DATASHEET

## E.1 MOTIVATION

- **For what purpose was the dataset created?** The dataset was created to address the current limitations of a lack of high-fidelity training data for the development and testing of machine learning methods for Computational Fluid Dynamics.

- **Who created the dataset (e.g., which team, research group) and on behalf of which entity (e.g., company, institution, organization)?** The dataset was created by a team of developers and scientists within the Advanced Computing & Simulation organisation of Amazon Web Services.

- **Who funded the creation of the dataset?** The project was internally funded within Amazon Web Services i.e no external grants.

## E.2 DISTRIBUTION

- **Will the dataset be distributed to third parties outside of the entity (e.g., company, institution, organization) on behalf of which the dataset was created?** Yes, the dataset is open to the public

- **How will the dataset will be distributed (e.g., tarball on website, API, GitHub)?** The dataset will be free to download from Amazon S3 (without the need for an AWS account).

- **When will the dataset be distributed?** The dataset is already available to download via Amazon S3.

- **Will the dataset be distributed under a copyright or other intellectual property (IP) license, and/or under applicable terms of use (ToU)?** The dataset is licensed under CC-BY-SA license.

- **Have any third parties imposed IP-based or other restrictions on the data associated with the instances?** No

- **Do any export controls or other regulatory restrictions apply to the dataset or to individual instances?** No

## E.3 MAINTENANCE

- **Who will be supporting/hosting/maintaining the dataset?** AWS are hosting the dataset on Amazon S3

- **How can the owner/curator/manager of the dataset be contacted (e.g., email address)?** The owner/curator/manager of the dataset can be contacted at xxxxx (these are also provided in the dataset README and paper).

- **Is there an erratum?** No, but if we find errors we will provide updates to the dataset and note any changes in the dataset README.

- **Will the dataset be updated (e.g., to correct labeling errors, add new instances, delete instances)?** Yes the dataset will be updated to address errors or provided extra functionality. The README of the dataset will be updated to reflect this.

- **If the dataset relates to people, are there applicable limits on the retention of the data associated with the instances (e.g., were the individuals in question told that their data would be retained for a fixed period of time and then deleted)?** N/A

- **Will older versions of the dataset continue to be supported/hosted/maintained?** Yes, if there are substantial changes or additions, older versions will still be kept.

- **If others want to extend/augment/build on/contribute to the dataset, is there a mechanism for them to do so?** We will consider this on a case by case basis and they can contact xxxxxxxx to discuss this further.

## E.4 COMPOSITION

- **What do the instances that comprise the dataset represent (e.g., documents, photos, people, countries)?** Computational Fluid Dynamic simulations

- **How many instances are there in total (of each type, if appropriate)?** Table 5 details the specific outputs that are contained in the dataset for each of the 500 geometric variations of the Ahmed car body.

- **Does the dataset contain all possible instances or is it a sample (not necessarily random) of instances from a larger set?** The dataset is complete collection of simulations run to date.

- **What data does each instance consist of?** Table 5 details the specific outputs that are contained in the dataset for each of the 500 geometric variations of the Ahmed car body.

- **Is there a label or target associated with each instance?** N/A

- **Is any information missing from individual instances?** No, the dataset is fully described.

- **Are relationships between individual instances made explicit (e.g., users' movie ratings, social network links)?** N/A

- **Are there recommended data splits (e.g., training, development/validation, testing)?** We don't provide explicit recommendations.

- **Are there any errors, sources of noise, or redundancies in the dataset?** The errors associated with the Computational Fluid Dynamics method is discussed in the validation section of the main paper and SI.

- **Is the dataset self-contained, or does it link to or otherwise rely on external resources (e.g., websites, tweets, other datasets)?** It is self-contained.

- **Does the dataset contain data that might be considered confidential (e.g., data that is protected by legal privilege or by doctor patient confidentiality, data that includes the content of individuals' non-public communications)?** The data is fully open-source and not considered confidential.

- **Does the dataset contain data that, if viewed directly, might be offensive, insulting, threatening, or might otherwise cause anxiety?** No

### E.5    COLLECTION PROCESS

- **How was the data associated with each instance acquired?** The data was obtained through Computational Fluid Dynamics (CFD) simulations and then post-processed to extract only the required quantities.

- **What mechanisms or procedures were used to collect the data (e.g., hardware apparatus or sensor, manual human curation, software program, software APIs?** The main paper discusses the HPC hardware used

- **If the dataset is a sample from a larger set, what was the sampling strategy (e.g., deterministic, probabilistic with specific sampling probabilities)?** N/A

- **Who was involved in the data collection process (e.g., students, crowdworkers, contractors) and how were they compensated (e.g., how much were crowdworkers paid)?** N/A

- **Over what timeframe was the data collected?** Simulation were run over the year of 2024.

- **Were any ethical review processes conducted (e.g., by an institutional review board)?** N/A

### E.6    PREPROCESSING/CLEANING/LABELING

- **Was any preprocessing/cleaning/labeling of the data done (e.g., discretization or bucketing, tokenization, part-of-speech tagging, SIFT feature extraction, removal of instances, processing of missing values)?** N/A

### E.7    USES

- **Has the dataset been used for any tasks already?** Yes, limited testing with various ML approaches has been undertaken by the author team to ensure that the data provided in the dataset is suitable for ML training and inference.

- **Is there a repository that links to any or all papers or systems that use the dataset?** No

- **What (other) tasks could the dataset be used for?** The primary focus is for ML development and testing but it could also be used for the study of turbulent flows over bluff bodies.

- **Is there anything about the composition of the dataset or the way it was collected and preprocessed/cleaned/labeled that might impact future uses?** Not to the knowledge of the authors.

- **Are there tasks for which the dataset should not be used?** No

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
