# OpenReview forum: "AhmedML: High-Fidelity Computational Fluid Dynamics Dataset for Incompressible, Low-Speed Bluff Body Aerodynamics"
_ICLR.cc/2025/Conference — Submitted to ICLR 2025_

### Official Review · Reviewer_CqRX · 2024-11-03

**Soundness:** 3
**Presentation:** 3
**Contribution:** 3
**Rating:** 6
**Confidence:** 5

**Summary:**

The paper introduces "AhmedML," a high-fidelity, open-source dataset for computational fluid dynamics (CFD) simulations focused on bluff body aerodynamics. The dataset includes 500 variations of the Ahmed car body, capturing crucial aerodynamic features such as 3D vortical structures and flow separation. Generated using a hybrid RANS-LES approach in OpenFOAM, the dataset is comprehensive, containing boundary and volume data, forces, moments, and STL files for geometry, with a reproducible case setup provided. This dataset fills a critical gap in accessible, large-scale CFD data for machine learning (ML) model development, aiming to support advancements in CFD and surrogate modeling for automotive applications.

**Strengths:**

+ High fidelity of the dataset ensures that its simulation data closely matches experimental results, achieved through advanced hybrid RANS-LES turbulence modeling.
+ The dataset enables ML models to be tested and trained on a wide range of flow conditions and physical characteristics.
+ The open-source nature and permissive license (CC-BY-SA) make the dataset widely accessible.
+ Providing a complete OpenFOAM case setup facilitates reproducibility, allowing other researchers to verify results or even expand the dataset, which supports a consistent foundation for model benchmarking.

**Weaknesses:**

- The dataset’s limitation to a single set of boundary conditions restricts its application to flow scenarios typical of the Ahmed body geometry.
- Although the Ahmed body is a well-known aerodynamic benchmark, it is a simplified model and lacks the intricate features of actual vehicle geometries, reducing generalizability to objects with sharp intricate features.
- The absence of time-resolved data means users are restricted to time-averaged flow characteristics, which may limit the use of the dataset in applications that require dynamic flow information, such as transient or unsteady flow modeling.

**Questions:**

1. How does this dataset compare against other open-source datasets for flow modeing?
2. How much more data will will it take to store the time series data?

---

### Official Review · Reviewer_7eJs · 2024-11-03

**Soundness:** 1
**Presentation:** 1
**Contribution:** 2
**Rating:** 3
**Confidence:** 3

**Summary:**

The paper introduces AhmedML, an open-source, high-fidelity computational fluid dynamics dataset designed to advance machine learning applications in aerodynamic simulations. The dataset features 500 geometric variations of the Ahmed Car Body, a well-known simplified vehicle shape used to study bluff body aerodynamics. To generate these simulations, the authors utilize a hybrid RANS-LES turbulence modeling approach within the OpenFOAM framework, ensuring accurate flow features that are typical in practical aerodynamic scenarios. AhmedML includes an extensive set of data, such as boundary conditions, volume flow fields, geometric properties, and time-averaged forces and moments, all formatted for easy use in ML training and validation.

**Strengths:**

A key strength lies in the use of advanced turbulence modeling techniques. The hybrid RANS-LES approach offers a better representation of complex flow features, such as vortex structures and flow separation, compared to lower-fidelity methods like steady-state RANS. This makes the dataset valuable for training machine learning models that require realistic aerodynamic data. The dataset includes not only the boundary and volume flow fields but also force and moment coefficients, making it suitable for a wide range of ML tasks. Additionally, the use of parametric variations of the Ahmed Car Body allows for robust testing of ML models' generalization capabilities across different geometric configurations. The authors have also made efforts to ensure that the data formats are accessible, which will facilitate adoption by both academic and industry researchers.

**Weaknesses:**

1. The use of time-averaged data instead of time-series data is a limitation. While time-averaged quantities are useful for certain engineering applications, the absence of transient data prevents the development of ML models that can predict unsteady aerodynamic behaviors. The inclusion of some runs with full time-history could enhance the dataset’s utility for dynamic flow predictions.
2. The Ahmed Car Body, though a standard benchmark for bluff body aerodynamics, does not capture the complexity of real-world vehicle geometries. This simplification may limit the dataset’s relevance for industrial applications.
3. The grammar in the manuscript is consistently poor, with frequent errors like missing articles, incorrect verb tenses, and awkward sentence structures. These issues make the paper difficult to read.

**Questions:**

Please see Weaknesses.

---

### Official Review · Reviewer_XdHq · 2024-11-03

**Soundness:** 3
**Presentation:** 3
**Contribution:** 2
**Rating:** 3
**Confidence:** 5

**Summary:**

This paper presents a new dataset for use in Computational Fluid Dynamics (CFD), geared towards ML calculations. The dataset covers incompressible, low speed, bluff body aerodynamics, of the type arising in car bodies. Data was generated through the OpenFoam software package running Reynolds Averaged Navier Stokes (RANS) and Large Eddy Simulations (LES) approaches. The dataset uses an unstructured grid generated through the OpenFoam mesh generation utility, and varies various aspects of the bodies in consideration, such as the geometry (length, width, height) and angle of the rear window. Velocities and pressure profiles are provided, together with lift and drag coefficients. In other words, material is provided to conduct bluff body simulations using machine learning techniques.

They provide a single evaluation case using graph neural networks, as regards ML validation.

**Strengths:**

- CFD datasets are a welcome addition for ML modelling, and to this end, works of this type is essential
- The paper does a good job explaining the various pieces of the contribution. The methods are explained quite clearly.

**Weaknesses:**

- Only a single evaluation using graph neural networks is provided, and even that is stated as being a work in progress.
"Please note that these runs are preliminary and further work to optimize the
methodology and hyperparameters is on-going which will published in future papers."
- I take issue with the limited evaluations and ML case studies conducted in the paper. We need to run more experiments to evaluate the fitness of the dataset. Examples:
 a. Are the number of samples in a given setting enough to obtain reasonable numbers?
b. How does the dataset fare for a given task and method (surrogate modelling, for instance)?

- Comparison with the other datasets mentioned in the paper: WindsorML and DrivAerML. The paper says that the current work addresses the limitations of the previous datasets by conducting hybrid RANS/LES calculations. A bit more explanation would be helpful in differentiating between these works.

**Questions:**

See weaknesses above. I would like, at the very least, more treatment of the evaluations, and an extensive description of the case considered with GNNs. Especially, it would be helpful to show how the various applications mentioned in the paper (e.g. surrogate modelling) are benefitted by the dataset. The best way to show that would be to conduct some experiments on the dataset with some well known reference methods (e.g. PINNs).

---

> ### Comment · Reviewer_XdHq · 2024-11-26
> **Rebuttal**
>
> It does not appear that we have author rebuttals here. Any thoughts on the paper?

---

### Official Review · Reviewer_ACVi · 2024-11-03

**Soundness:** 3
**Presentation:** 3
**Contribution:** 1
**Rating:** 3
**Confidence:** 4

**Summary:**

This is a domain-specific dataset paper.  The authors produce a dataset consisting of various OpenFOAM simulations of fluid flow past 500 geometric variations of the Ahmed Car Body geometry.  The intent is to produce a dataset that is representative of the types of flows past various road vehicle bodies.  The authors use a hybrid RANS-LES solver within OpenFOAM.

**Strengths:**

1. I believe the introduction and related work are relatively well-written and cited.  One area of research the authors may want to cite is the growing body of work on using DL to improve the linear solvers used in CFD solvers, including neural preconditioners; for example, "A deep conjugate direction method for iteratively solving linear systems" (ICML 2023) and "A Neural-preconditioned Poisson Solver for Mixed Dirichlet and Neumann Boundary Conditions" (ICML 2024).

2. The dataset that is developed in this submission is licensed very permissively, which means anyone could use the results from this manuscript.

**Weaknesses:**

1. Although I do like the related work, an issue is that the authors state "several groups have generated large-scale training data to demonstrate the potential of their ML methods," and that the main issue is those datasets aren't free/open-source.  I do appreciate the value of an open-source dataset, but the novelty of the present work is pretty limited; the main advantage is that it's open-source.

2. Specifically, the difference between DrivAerNet and the present work appears to be (1) the specific geometries tested (this is easy to swap out), (2) the solver choice in OpenFOAM (this is an easy setting to change), and the license (DrivAerNet uses an open-source license, too, but not one the authors like).  While I appreciate that the present dataset's license allows for commercial usage, unlike DrivAerNet, I don't think this is a research contribution.

3. The authors emphasize the realism of their results, using somewhat accurate CFD algorithms in OpenFOAM, yet their "car" geometries are extremely basic.  While it's nice that their geometries are parametric and one can easily generate variations of these, if the emphasis of the paper is on realism, it seems like one should use a bunch of real-world (or even slightly more car-shaped) car geometries / CAD models.

4. Relatedly, the authors claim that using their chosen hybrid RANS-LES solver in OpenFOAM produces results that are more accurate than other models, but they do not demonstrate that - in the end - their dataset enables better results than a similar dataset crafted with the same geometries but a cheaper solver.  In other words: how much is gained by using this particular OpenFOAM solver?

5. The appendix/supplementary material for this submission seems exceptionally long.  The ICLR page limit is 9 pages, and the current submission extends to 41 pages.  I do not think this entire appendix is necessary in a submission.  For instance, Fig. 10 is a repeat of Fig. 6, and various text around there is repeated from the text in the main body.

6. (Minor) It would be nice to include the R^2 coefficients for Figure 6 or some similar statistical quantitative measurements of how good the fits are.

**Questions:**

I have no questions about the manuscript at this time.

**Details Of Ethics Concerns:**

The paper mentions how the authors come from the "Advanced Computing & Simulation organisation of Amazon Web Services."  While I don't know the authors' specific identities, this narrows it down enough where I worry reviewers might have unconscious biases.  I don't think it's a serious enough issue to reject the paper on its own, but I think the ACs should be aware in making their final decisions on this submission.

---

### Official Review · Reviewer_DzkF · 2024-11-05

**Soundness:** 3
**Presentation:** 3
**Contribution:** 2
**Rating:** 5
**Confidence:** 4

**Summary:**

The ML-based methods depends on the "true" solutions simulated by traditional CFD solvers. The high-quality dataset is lacking in the field of ML for CFD.

To address this issue, the paper uses higher-fidelity scale-resolving methods to generate dataset the ML training dataset for the widely used Ahmed car body, and make it freely available. The paper lists details of the dataset, and gives an example of using the proposed dataset to train a GNN model.

**Strengths:**

- The Ahmed car body dataset is quite comprehensive - it contains volume, boundary, geometry and forces & moments in
open-source formats
- The paper lists details about how to use the dataset, and makes it freely available.
- The paper also lists details of how the dataset was generated (such as the boundary conditions, HPC setups), which is good.

**Weaknesses:**

- The proposed dataset appears solid. However, I have some minor reservations regarding the paper’s contribution, as its primary contribution is introducing a dataset for a single type of shape.

**Questions:**

- Is the dataset somewhat limited, given that it only includes car-like shapes?
- For those interested in generating data for different shapes, how easily can this setup be adapted, compared to setting up a CFD simulation from scratch (utilizing the existing open-source simulation frameworks)?
- How much impact would training the model on a lower-quality or pre-existing dataset have on model quality compared to using the proposed dataset?
- Is the primary challenge in dataset generation related to configuring the OpenFOAM simulation or getting the computational resources needed to complete all simulations? or both? or others?

---

> ### Comment · Reviewer_DzkF · 2024-11-28
>
> I keep my original score as we don't have author rebuttals for this paper.

---

> > ### Author Response · Authors · 2024-11-28
> > **Response**
> >
> > Please accept our apologies for the delay in responding.
> >
> > On your specific points:
> >
> > Question 1: Is the dataset somewhat limited, given that it only includes car-like shapes?
> > Answer 1: whilst it is true that having a dataset containing many different type of geometries would be ideal, the dataset we have created is already first of a kind - there are no public datasets for this complexity using these high-fidelity scale-resolving approaches. So only having car-like shapes is in our view an acceptable compromise and makes it domain specific i.e automotive/bluff-body.
> >
> > Question 2: For those interested in generating data for different shapes, how easily can this setup be adapted, compared to setting up a CFD simulation from scratch (utilizing the existing open-source simulation frameworks)?
> > Answer 2: the dataset contains the complete simulation setup so people could easily reproduce the dataset or add additional geometries (all fully open-source)
> >
> > Question 3: How much impact would training the model on a lower-quality or pre-existing dataset have on model quality compared to using the proposed dataset?
> > Answer 3 : good question! We did not investigate this but we hope that this dataset being public will allow others to test this out.
> >
> > Question 4: Is the primary challenge in dataset generation related to configuring the OpenFOAM simulation or getting the computational resources needed to complete all simulations? or both? or others?
> > Answer 4: Is it a combination of all of this - which is why this is the free-ever work (in our knowledge) that has been made available publicly. As the paper points out, creating this dataset required high-fidelity CFD knowledge and validation work, as well as the HPC/compute knowledge to create this efficiently.
> >
> > We respectually would like to point out that your comments are 'minor' by your own wording and therefore we would say that a rating of 5 i.e below acceptable threshold is not reflecting your actual weaknesses/strengths. we understand with minor issues you would not want to give an 8 but 5 does not seem to reflect what you are pointing out as minor weaknesses and therefore we believe a higher score is warranted.

---

### Comment · Area_Chair_1SNV · 2024-11-25
**Authors' Rebuttal**

Dear Authors,

As the author-reviewer discussion period is approaching its end, I strongly encourage you to read the reviews and engage with the reviewers to ensure the message of your paper has been appropriately conveyed and any outstanding questions have been resolved.

This is a crucial step, as it ensures that both reviewers and authors are on the same page regarding the paper's strengths and areas for improvement.

Thank you again for your submission.

Best regards,

AC

---

### Meta-Review · Area_Chair_1SNV · 2024-12-17

**Metareview:**

This domain-specific dataset paper introduces AhmedML, a high-fidelity, open-source dataset for CFD designed to advance ML applications in aerodynamic simulations. The dataset provides a collection of simulations using a hybrid RANS-LES approach in OpenFOAM for 500 geometric variations of a car body, a standard benchmark for bluff body aerodynamics.

The paper is well-written with clear details on the dataset creation process and its very permissive license. However, the paper presents simulations based on only one car body geometry, which one reviewer argues does not capture real-world complexities. Additionally, it provides only a simple Graph Neural Network (GNN) model with limited evaluation and no comprehensive ML baselines for comparison.

As most reviewers' comments agree on the very-limited number of configurations, which do not capture the complexity of real car design, and the lack of ML baselines, I recommend rejection.

**Additional Comments On Reviewer Discussion:**

The responses from the authors were quite limited. No modification of the manuscript.

---

### Decision · Program_Chairs · 2025-01-22

Reject